# DIFFERENTIABLE HEBBIAN CONSOLIDATION FOR CONTINUAL LEARNING

## ABSTRACT

Continual learning is the problem of sequentially learning new tasks or knowledge while protecting previously acquired knowledge. However, catastrophic forgetting poses a grand challenge for neural networks performing such learning process. Thus, neural networks that are deployed in the real world often struggle in scenarios where the data distribution is non-stationary (concept drift), imbalanced, or not always fully available, i.e., rare edge cases. We propose a Differentiable Hebbian Consolidation model which is composed of a Differentiable Hebbian Plasticity (DHP) Softmax layer that adds a rapid learning plastic component (compressed episodic memory) to the fixed (slow changing) parameters of the softmax output layer; enabling learned representations to be retained for a longer timescale. We demonstrate the flexibility of our method by integrating well-known task-specific synaptic consolidation methods to penalize changes in the slow weights that are important for each target task. We evaluate our approach on the Permuted MNIST, Split MNIST and Vision Datasets Mixture benchmarks, and introduce an imbalanced variant of Permuted MNIST — a dataset that combines the challenges of class imbalance and concept drift. Our proposed model requires no additional hyperparameters and outperforms comparable baselines by reducing forgetting.

## 1 INTRODUCTION

A key aspect of human intelligence is the *ability to continually adapt and learn* in dynamic environments, a characteristic which is challenging to embed into artificial intelligence. Recent advances in machine learning (ML) have shown tremendous improvements in various problems, by learning to solve one complex task very well, through extensive training on large datasets with millions of training examples or more. However, most of the ML models that are used during deployment in the real-world are exposed to non-stationarity where the distributions of acquired data changes over time. Therefore, after learning is complete, and these models are further trained with new data, responding to distributional changes, performance degrades with respect to the original data. This phenomenon known as *catastrophic forgetting* or *catastrophic interference* (McCloskey & Cohen, 1989; French, 1999) presents a crucial problem for deep neural networks (DNNs) that are tasked with continual learning (Ring, 1994), also called lifelong learning (Thrun & Mitchell, 1995; Thrun, 1998). In continual learning, the goal is to adapt and learn consecutive tasks without forgetting how to perform well on previously learned tasks, enabling models that are scalable and efficient over long timescales.

In most supervised learning methods, DNN architectures require independent and identically distributed (iid) samples from a stationary training distribution. However, for ML systems in real-world applications that require continual learning, the iid assumption is easily violated when: (1) There is concept drift in the training data distribution. (2) There are imbalanced class distributions and concept drift occuring simultaneously. (3) Data representing all scenarios in which the learner is expected to perform are not initially available. In such situations, learning systems face the "stability-plasticity dilemma" which is a well-known problem for artificial and biological neural networks (Carpenter & Grossberg, 1987; Abraham & Robins, 2005). This presents a continual learning challenge for an ML system where the model needs to provide a balance between its plasticity (to integrate new knowledge) and stability (to preserve existing knowledge).

In biological neural networks, synaptic plasticity has been argued to play an important role in learning and memory (Howland & Wang, 2008; Takeuchi et al., 2013; Bailey et al., 2015) and two major

theories have been proposed to explain a human's ability to perform continual learning. The first theory is inspired by synaptic consolidation in the mammalian neocortex (Benna & Fusi, 2016) where a subset of synapses are rendered less plastic and therefore preserved for a longer timescale. The general idea for this approach is to consolidate and preserve synaptic parameters that are considered important for the previously learned tasks. This is normally achieved through task-specific updates of synaptic weights in a neural network. The second is the complementary learning system (CLS) theory (McClelland et al., 1995; Kumaran et al., 2016), which suggests that humans extract high-level structural information and store it in different brain areas while retaining episodic memories.

Recent work on differentiable plasticity has shown that neural networks with "fast weights" that leverage Hebbian learning rules (Hebb, 1949) can be trained end-to-end through backpropagation and stochastic gradient descent (SGD) to optimize the standard "slow weights", as well as also the amount of plasticity in each synaptic connection (Miconi, 2016; Miconi et al., 2018). These works use slow weights to refer to the weights normally used to train vanilla neural networks, which are updated slowly and are often associated with long-term memory. The fast weights represent the weights that are superimposed on the slow weights and change quickly from one time step to the next based on input representations. These fast weights behave as a form of short-term memory that enable "reactivation" of long-term memory traces in the slow weights. Miconi et al. (2018) showed that simple plastic networks with learned plasticity outperform networks with uniform plasticity on various problems. Moreover, there have been several approaches proposed recently for overcoming the catastrophic forgetting problem in fixed-capacity models by dynamically adjusting the plasticity of each synapse based on its importance for retaining past memories (Parisi et al., 2019).

Here, we extend the work on differentiable plasticity to the *task-incremental* continual learning setting (van de Ven & Tolias, 2019), where tasks arrive in a batch-like fashion, and have clear boundaries. We develop a Differentiable Hebbian Consolidation[1] model that is capable of adapting quickly to changing environments as well as consolidating previous knowledge by selectively adjusting the plasticity of synapses. We modify the traditional softmax layer and propose to augment the slow weights in the final fully-connected (FC) layer (softmax output layer) with a set of plastic weights implemented using Differentiable Hebbian Plasticity (DHP). Furthermore, we demonstrate the flexibility of our model by combining it with recent task-specific synaptic consolidation based approaches to overcoming catastrophic forgetting such as elastic weight consolidation (Kirkpatrick et al., 2017; Schwarz et al., 2018), synaptic intelligence (Zenke et al., 2017b) and memory aware synapses (Aljundi et al., 2018). Our model unifies core concepts from Hebbian plasticity, synaptic consolidation and CLS theory to enable rapid adaptation to new unseen data, while consolidating synapses and leveraging compressed episodic memories in the softmax layer to remember previous knowledge and mitigate catastrophic forgetting. We test our proposed method on established benchmark problems including the Permuted MNIST (Goodfellow et al., 2013), Split MNIST (Zenke et al., 2017b) and Vision Datasets Mixture (Ritter et al., 2018) benchmarks. We also introduce the Imbalanced Permuted MNIST problem and show that plastic networks with task-specific synaptic consolidation methods outperform networks with uniform plasticity.

## 2 RELEVANT WORK

**Neural Networks with Non-Uniform Plasticity:** One of the major theories that have been proposed to explain a human's ability to learn continually is Hebbian learning (Hebb, 1949), which suggests that learning and memory are attributed to weight plasticity, that is, the modification of the strength of existing synapses according to variants of Hebb's rule (Paulsen & Sejnowski, 2000; Song et al., 2000; Oja, 2008). It is a form of activity-dependent synaptic plasticity where correlated activation of pre- and post-synaptic neurons leads to the strengthening of the connection between the two neurons. According to the Hebbian learning theory, after learning, the related synaptic strength are enhanced while the degree of plasticity decreases to protect the learned knowledge (Zenke et al., 2017a).

Recent approaches in the meta-learning literature have shown that we can incorporate fast weights into a neural network to perform one-shot and few-shot learning (Munkhdalai & Trischler, 2018; Rae et al., 2018). Munkhdalai & Trischler (2018) proposed a model that augments FC layers preceding the softmax with a matrix of fast weights to bind labels to representations. Here, the fast weights were implemented with *non-trainable* Hebbian learning-based associative memory. Rae et al. (2018)

---

[1]Code is available at:

proposed a Hebbian Softmax layer that can improve learning of rare classes by interpolating between Hebbian learning and SGD updates on the output layer using an engineered scheduling scheme.

Miconi et al. (2018) proposed differentiable plasticity, which uses SGD to optimize the plasticity of each synaptic connection, in addition to the standard fixed (slow) weights. Here, each synapse is composed of a slow weight and a plastic (fast) weight that automatically increases or decreases based on the activity over time. Although this approach served to be a powerful new method for training neural networks, it was mainly demonstrated on recurrent neural networks (RNNs) for solving pattern memorization tasks and maze exploration with reinforcement learning. Also, these approaches were only demonstrated on meta-learning problems and not the continual learning challenge of overcoming catastrophic forgetting. Our work also augments the slow weights in the FC layer with a set of plastic (fast) weights, but implements these using DHP. We only update the parameters of the softmax output layer in order to achieve fast learning and preserve knowledge over time.

**Overcoming Catastrophic Forgetting:** This work leverages two strategies to overcome the catastrophic forgetting problem: 1) *Task-specific Synaptic Consolidation* — Protecting previously learned knowledge by dynamically adjusting the synaptic strengths to consolidate and retain memories. 2) *CLS Theory* — A dual memory system where, the neocortex (neural network) gradually learns to extract structured representations from the data while, the hippocampus (augmented episodic memory) performs rapid learning and individuated storage to memorize new instances or experiences.

There have been several notable works inspired by task-specific synaptic consolidation for overcoming catastrophic forgetting (Kirkpatrick et al., 2017; Zenke et al., 2017b; Aljundi et al., 2018) and they are often categorized as regularization strategies in the continual learning literature (Parisi et al., 2019). All of these regularization approaches estimate the importance of each parameter or synapse, $\Omega_k$, where least plastic synapses can retain memories for long timescales and more plastic synapses are considered less important. The parameter importance and network parameters $\theta_k$ are updated in either an online manner or after learning task $T_n$. Therefore, when learning new task $T_{n+1}$, a regularizer is added to the original loss function $\mathcal{L}^n(\theta)$, so that we dynamically adjust the plasticity w.r.t $\Omega_k$ and prevent any changes to important parameters of previously learned tasks:

$$\tilde{\mathcal{L}}^n(\theta) = \mathcal{L}^n(\theta) + \lambda \underbrace{\sum_k \Omega_k (\theta_k^n - \theta_k^{n-1})^2}_{\text{regularizer}} \tag{1}$$

where $\theta_k^{n-1}$ are the learned network parameters after training on the previous $n-1$ tasks and $\lambda$ is a hyperparameter for the regularizer to control the amount of forgetting (old versus new memories).

The main difference in these regularization strategies is on the method used to compute the importance of each parameter, $\Omega_k$. In Elastic Weight Consolidation (EWC), Kirkpatrick et al. (2017) used the values given by the diagonal of an approximated Fisher information matrix for $\Omega_k$, and this was computed offline after training on a task was completed. An online variant of EWC was proposed by Schwarz et al. (2018) to improve EWC's scalability by ensuring the computational cost of the regularization term does not grow with the number of tasks. Zenke et al. (2017b) proposed an online method called Synaptic Intelligence (SI) for computing the parameter importance where, $\Omega_k$ is the cumulative change in individual synapses over the entire training trajectory on a particular task. Memory Aware Synapses (MAS) from Aljundi et al. (2018) is an online method that measures $\Omega_k$ by the sensitivity of the learned function to a perturbation in the parameters, instead of measuring the change in parameters to the loss as seen in SI and EWC.

Our work draws inspiration from CLS theory which is a powerful computational framework for representing memories with a dual memory system via the neocortex and hippocampus. There have been numerous approaches based on CLS principles involving pseudo-rehersal (Robins, 1995; Ans et al., 2004; Atkinson et al., 2018), exact or episodic replay (Lopez-Paz & Ranzato, 2017; Li & Hoiem, 2018) and generative replay (Shin et al., 2017; Wu et al., 2018). Exact replay methods require storage of the data from previous tasks which are later replayed. Generative replay methods train a separate generative model to generate images to be replayed. iCaRL (Rebuffi et al., 2017) performs rehearsal and regularization, where an external memory is used to store exemplar patterns from old task data and rehearse the model via distillation. However, in our work, we are primarily interested in neuroplasticity techniques inspired from CLS theory for alleviating catastrophic forgetting. Earlier work from Hinton & Plaut (1987); Gardner-Medwin (1989) showed how each synaptic connection can be composed of a fixed weight where slow learning stores long-term knowledge and

a fast-changing weight for temporary associative memory. This approach involving slow and fast weights is analogous to properties of CLS theory to overcome catastrophic forgetting during continual learning. Recent research in this vein has included replacing soft attention mechanism with fast weights in RNNs (Ba et al., 2016), the Hebbian Softmax layer (Rae et al., 2018), augmenting slow weights in the FC layer with a fast weights matrix (Munkhdalai & Trischler, 2018), differentiable plasticity (Miconi, 2016; Miconi et al., 2018) and neuromodulated differentiable plasticity (Miconi et al., 2019).

We did not evaluate and compare against neuroplasticity-inpired CLS methods as baselines because they were designed for meta-learning problems and would be unfair to evaluate their performance on continual learning benchmark problems given some of their limitations. All of these methods were designed for rapid learning on simple tasks or meta-learning over a distribution of tasks or datasets, where a few number of examples from a class are seen by the network when training on different tasks to perform one-shot and few-shot learning. For instance, the Hebbian Softmax layer modifies its parameters by annealing between Hebbian and SGD updates based on an engineered scheduling scheme which achieves fast binding for rarer classes. However, when a large number of examples are observed frequently from the same class, the annealing function switches completely to SGD updates. Thus, when evaluating this model in continual learning setups, the effect of the fast weights memory storage becomes non-existent as the network learns from a large number of examples per class on each task. With a focus on continual learning, the goal of our work is to metalearn a local learning rule for the fast weights via the fixed (slow) weights and an SGD optimizer.

## 3 DIFFERENTIABLE HEBBIAN CONSOLIDATION

In our model, each synaptic connection in the softmax layer has two weights: 1) The slow weights, $\theta \in \mathbb{R}^{m \times d}$, where $m$ is the number of units in the final hidden layer and $d$ is the number of outputs of the last layer. 2) A Hebbian plastic component of the same cardinality as the slow weights, composed of the plasticity coefficient, $\alpha$, and the Hebbian trace, Hebb. The $\alpha$ is a scaling parameter for adjusting the magnitude of the Hebb. The Hebbian traces accumulate the mean hidden activations of the final hidden layer $h$ for each target label in the mini-batch $\{y_{1:B}\}$ of size $B$ which are denoted by $\tilde{h} \in \mathbb{R}^{1 \times m}$ (refer to Algorithm 1). Given the pre-synaptic activations of neurons $i$ in $h$, we can formally compute the post-synaptic activations of neurons $j$ using Eq. 2 and obtain the unnormalized log probabilities (softmax pre-activations) $z$. The softmax function is then applied on $z$ to obtain the desired predicted probabilities $\hat{y}$ thus, $\hat{y} = \text{softmax}(z)$. The $\eta$ parameter in Eq. 3 is a scalar value that dynamically learns how quickly to acquire new experiences into the plastic component, and thus behaves as the learning rate for the plastic connections. The $\eta$ parameter also acts as a decay term for the Hebb to prevent instability caused by a positive feedback loop in the Hebbian traces.

$$z_j = \sum_{i=1}^{m} (\underbrace{\theta_{i,j}}_{\text{slow}} + \underbrace{\alpha_{i,j}\text{Hebb}_{i,j}}_{\text{plastic (fast)}})h_i \tag{2}$$

$$\text{Hebb}_{i,j} \leftarrow (1 - \eta)\text{Hebb}_{i,j} + \eta\tilde{h}_{i,j} \tag{3}$$

The network parameters $\alpha_{i,j}$, $\eta$ and $\theta_{i,j}$ are optimized by gradient descent as the model is trained sequentially on different tasks in the continual learning setup. In standard neural networks the weight connection has only fixed (slow) weights, which is equivalent to setting the plasticity coefficients $\alpha = 0$ in Eq. 2.

**Hebbian Update Rule:** The Hebbian traces are initialized to zero only at the start of learning the first task $T_1$ and during training, the Hebb is automatically updated in the forward pass using Algorithm 1. Specifically, the Hebbian update for a coressponding class $c$ in $y_{1:B}$ is computed on line 6. This Hebbian update $\frac{1}{s}\sum_{b=1}^{B} h[y_b = c]$ is analogous to another formulaic description of the Hebbian learning update rule

---

**Algorithm 1** Batch update Hebbian traces.

1: **Input:** $h_{1:B}$ (hidden activations of penultimate layer),
    $y_{1:B}$ (target labels),
    Hebb (Hebbian trace)
2: **Output:** $z_{1:B}$ (softmax pre-activations)
3: **for** each target label $c \in \{y_{1:B}\}$ **do**
4:      $s \leftarrow \sum_{b=1}^{B}[y_b = c]$     /*Count total occurences of $c \in y$.*/
5:      **if** $s > 0$ **then**
6:         $\tilde{h} \leftarrow \frac{1}{s}\sum_{b=1}^{B} h[y_b = c]$    /*Update Hebb for class c.*/
7:         $\text{Hebb}_{:,c} \leftarrow (1 - \eta)\text{Hebb}_{:,c} + \eta\tilde{h}$
8:      **end if**
9: **end for**
10: $z \leftarrow (\theta + \alpha\text{Hebb})h$     /*Compute softmax pre-activations.*/

---

$w_{i,j} = \frac{1}{N} \sum_{k=1}^{N} a_i^k a_j^k$ (Hebb, 1949), where $w_{i,j}$ is the change in weight at connection $i, j$ and $a_i^k, a_j^k$ denote the activation levels of neurons $i$ and $j$, respectively, for the $k^{\text{th}}$ input. Therefore, in our model, $w = \tilde{h}$ the Hebbian weight update, $a_i = h$ the hidden activations of the last hidden layer, $a_j = y$ the corresponding target class in $y_{1:B}$ and $N = s$ the number of inputs for the corresponding class in $y_{1:B}$ (see Algorithm 1). Across the model's lifetime, we only update the Hebbian traces during training as it learns tasks in a continual manner. Therefore, during test time, we maintain and use the most recent $\text{Hebb}$ traces to make predictions.

Our model explores an optimization scheme where hidden activations are accumulated directly into the softmax output layer weights when a class has been seen by the network. This results in better initial representations and can also retain these learned deep representations for a much longer timescale. This is because memorized activations for one class are not competing for space with activations from other classes. Fast learning, enabled by a highly plastic weight component, improves test accuracy for a given task. Between tasks this plastic component decays to prevent interference, but selective consolidation into a stable component protects old memories, effectively enabling the model to *learn to remember* by modelling plasticity over a range of timescales to form a learned neural memory (see Section 4.1 ablation study). In comparison to an external memory, the advantage of DHP Softmax is that it is simple to implement, requiring no additional space or computation. This allows it to scale easily with increasing number of tasks.

The plastic component learns rapidly and performs sparse parameter updates to quickly store memory traces for each recent experience without interference from other similar recent experiences. Furthermore, the hidden activations corresponding to the same class, $c$, are accumulated into one vector $\tilde{h}$, thus forming a compressed episodic memory in the Hebbian traces to reflect individual episodic memory traces (similar to the hippocampus in biological neural networks (Chadwick et al., 2010; Schapiro et al., 2017)). As a result, this method improves learning of rare classes and speeds up binding of class labels to deep representations of the data without introducing any additional hyperparameters. In Appendix B, we provide a sample implementation of the DHP Softmax using PyTorch.

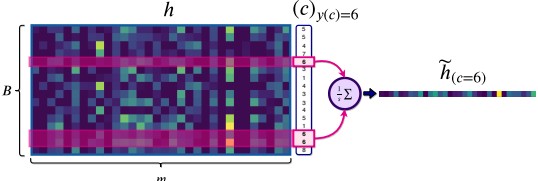

Figure 1: An example of a Hebbian update for the class, $c = 6 \in y_{1:B}$. Here, we are given the hidden activations of the final hidden layer, $h$. Multiple hidden activations corresponding to class $c = 6$ (represented by the pink boxes) are averaged into one vector denoted by $\tilde{h} \in \mathbb{R}^{1 \times m}$. This Hebbian update visualization reflects Lines 4-6 in Algorithm 1 and is repeated for each unique class in the target vector $y_{1:B}$.

**Hebbian Synaptic Consolidation:** Following the existing regularization strategies such as EWC (Kirkpatrick et al., 2017), Online EWC (Schwarz et al., 2018), SI (Zenke et al., 2017b) and MAS (Aljundi et al., 2018), we regularize the loss $\mathcal{L}(\theta)$ as in Eq. 1 and update the synaptic importance parameters of the network in an online manner. We rewrite Eq. 1 to obtain the updated quadratic loss for Hebbian Synaptic Consolidation in Eq. 4 and show that the network parameters $\theta_{i,j}$ are the weights of the connections between pre- and post-synaptic activities of neurons $i$ and $j$, as seen in Eq. 2.

$$\tilde{\mathcal{L}}^n(\theta, \alpha, \eta) = \mathcal{L}^n(\theta, \alpha, \eta) + \lambda \sum_{i,j} \Omega_{i,j}(\theta_{i,j}^n - \theta_{i,j}^{n-1})^2 \tag{4}$$

We adapt the existing task-specific consolidation approaches to our model and do not compute the synaptic importance parameters on the plastic component of the network, hence we only regularize the slow weights of the network. Furthermore, when training the first task $T_{n=1}$, the synaptic importance parameter, $\Omega_{i,j}$ in Eq. 4, was set to 0 for all of the task-specific consolidation methods that we tested on except for SI. This is because SI is the only method we evaluated that estimates $\Omega_{i,j}$ while training, whereas Online EWC and MAS compute $\Omega_{i,j}$ after learning a task. The plastic component of the softmax layer in our model can alleviate catastrophic forgetting of consolidated classes by allowing gradient descent to optimize how plastic the connections should be (i.e. less plastic to preserve old information or more plastic to quickly learn new information).

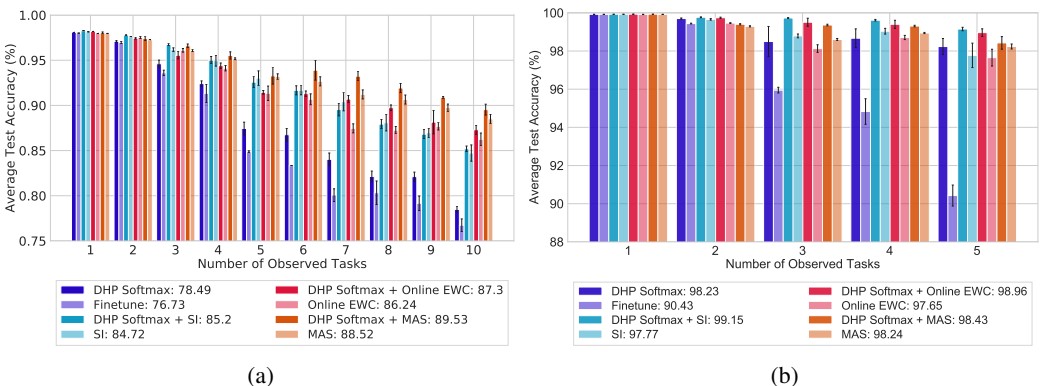

(a)                                                  (b)

Figure 2: (a) The average test accuracy on a sequence of 10 Permuted MNIST tasks $T_{n=1:10}$ and (b) a sequence of 5 binary classification tasks from the original MNIST dataset $T_{n=1:5}$. The average test accuracy over all learned tasks is provided in the legend. The addition of DHP in all cases improves the model's ability to reduce forgetting. The error bars correspond to the SEM across 10 trials.

## 4 EXPERIMENTS

In our experiments, we compare our approach to vanilla neural networks with Online EWC, SI and MAS. Since our approach increases the capacity of the DNN due to the addition of plastic weights, we add an extra set of slow weights to the softmax output layer of the standard neural network to match the capacity. We do this to show that it is not the increased model capacity from the plastic weights that is helping mitigate the forgetting when performing sequential task learning, thus ensuring a fair evaluation. We tested our model on the Permuted MNIST, Split MNIST and Vision Datasets Mixture benchmarks, and also introduce the Imbalanced Permuted MNIST problem.

For all of the benchmarks, we evaluated the model based on the average classification accuracy on all previously learned tasks as a function of $n$, the number of tasks trained so far. To determine memory retention and flexibility of the model, we are particularly interested in the test performance on the first task and the most recent one. We also measure forgetting using the backward transfer metric, $\text{BWT} = \frac{1}{T-1} \sum_{i=1}^{T-1} R_{T,i} - R_{i,i}$ (Lopez-Paz & Ranzato, 2017), which indicates how much learning new tasks has influenced the performance on previous tasks. $R_{T,i}$ is the test classification accuracy on task $i$ after sequentially finishing learning the $T^{\text{th}}$ task. While $\text{BWT} < 0$ directly reports catastrophic forgetting, $\text{BWT} > 0$ indicates that learning new tasks has helped with the preceding tasks. To establish a baseline for comparison of well-known task-specific consolidation methods, we trained neural networks with Online EWC, SI and MAS, respectively, on all tasks in a sequential manner. The hyperparameters of the consolidation methods (i.e. EWC, SI and MAS) remain the same with and without DHP Softmax, and the plastic components are not regularized. Descriptions of the hyperparameters and other details for all benchmarks can be found in Appendix A.

### 4.1 PERMUTED MNIST

In this benchmark, all of the MNIST pixels are permuted differently for each task with a fixed random permutation. Although the output domain is constant, the input distribution changes between tasks and is mostly independent of each other, thus, there exists a concept drift. In the Permuted MNIST and Imbalanced Permuted MNIST benchmarks we use a multi-layered perceptron (MLP) network with two hidden layers consisting of 400 ReLU nonlinearities, and a cross-entropy loss. The $\eta$ of the plastic component was set to be a value of 0.001 and we emphasize that we spent little to no effort on tuning the initial value of this parameter (see Appendix A.5 for a sensitivity analysis).

We first compare the performance between our network with DHP Softmax and a fine-tuned vanilla MLP network we refer to as *Finetune* in Figure 2a and no task-specific consolidation methods involved. The network with DHP Softmax alone showed improvement in its ability to alleviate catastrophic forgetting across all tasks compared to the baseline network. Then we compared the performance with and without DHP Softmax using the same task-specific consolidation methods.

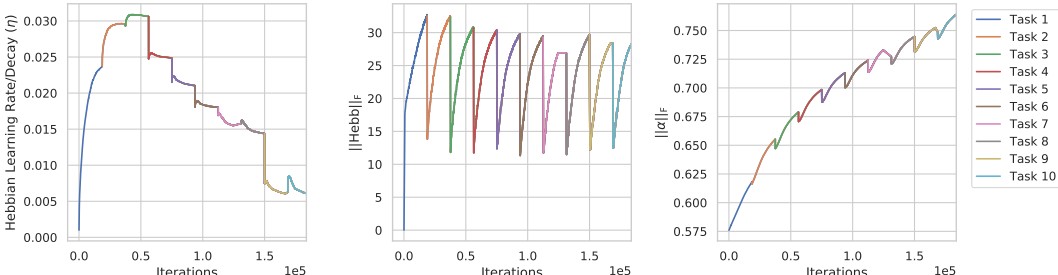

Figure 3: **(left)** Hebbian learning rate and decay value $\eta$, **(middle)** Frobenius Norm of the Hebbian memory traces $\|\text{Hebb}\|_F$, **(right)** Frobenius Norm of the plasticity coefficients $\|\alpha\|_F$ while training each task $T_{1:10}$.

Figure 2a shows the average test accuracy as new tasks are learned for the best hyperparameter combination for each task-specific consolidation method. We find our DHP Softmax with consolidation maintains a higher test accuracy throughout sequential training of tasks than without DHP Softmax.

**Ablation Study:** We further examine the structural parameters of the network and Hebb traces to provide further interpretability into the behaviour of our proposed model. The left plot in Figure 8 shows the behaviour of $\eta$ during training as 10 tasks in the Permuted MNIST benchmark are learned continually. Initially, in task $T_1$, $\eta$ increases very quickly from 0.001 to 0.024 suggesting that the synaptic connections become more plastic to quickly acquire new information. Eventually, $\eta$ decays after the $3^{\text{rd}}$ task to reduce the degree of plasticity to prevent interference between the learned representations. We also observe that within each task from $T_4$ to $T_{10}$, $\eta$ initially increases then decays. The Frobenius Norm of the Hebb trace (middle plot in Figure 8) suggests that Hebb grows *without* runaway positive feedback every time a new task is learned, maintaining a memory of which synapses contributed to recent activity. The Frobenius Norm of $\alpha$ (right plot in Figure 8) indicates that the plasticity coefficients grow within each task, indicating that the network is leveraging the structure in the plastic component. It is important to note that gradient descent and backpropagation are used as *meta-learning* to tune the structural parameters in the plastic component.

## 4.2 IMBALANCED PERMUTED MNIST

We introduce the Imbalanced Permuted MNIST problem which is identical to the Permuted MNIST benchmark but, now each task is an imbalanced distribution where training samples in each class were artificially removed based on some random probability (see Appendix A.2). This benchmark was motivated by the fact that class imbalance and concept drift can hinder predictive performance, and the problem becomes particularly challenging when they occur simultaneously. Appendix A.6, Figure 5 shows the average test accuracy for the best hyperparameters of each method. We see that DHP Softmax achieves 80.85% after learning 10 tasks with imbalanced class distributions in a sequential manner, thus providing significant 4.41% improvement over the standard neural network baseline of 76.44%. The significance of the compressed episodic memory mechanism in the Hebbian traces is more apparent in this benchmark because the plastic component allows rare classes that are encountered infrequently to be remembered for a longer period of time. We find that DHP Softmax with MAS achieves a 0.04 decrease in BWT, resulting in an average test accuracy of 88.80% and a 1.48% improvement over MAS alone; also outperforming all other methods and across all tasks.

## 4.3 SPLIT MNIST

We split the original MNIST dataset (LeCun et al., 2001) into a sequence of 5 binary classification tasks: $T_1 = \{0/1\}$, $T_2 = \{2/3\}$, $T_3 = \{4/5\}$, $T_4 = \{6/7\}$ and $T_5 = \{8/9\}$. The output spaces are disjoint between tasks, unlike the previous two benchmarks. Similar to the network used by Zenke et al. (2017b), we use an MLP network with two hidden layers of 256 ReLU nonlinearities each, and a cross-entropy loss. The initial $\eta$ value was set to 0.001 as seen in previous benchmark experiments. We found that different values of $\eta$ yielded very similar final test performance after learning $T_5$ tasks (see Appendix A.5). We observed that DHP Softmax alone achieves 98.23% thus, provides a 7.80% improvement on test performance compared to a finetuned MLP network (Figure 2b). Also,

combining DHP Softmax with task-specific consolidation consistently decreases BWT, leading to a higher average test accuracy across all tasks, especially the most recent one, $T_5$.

## 4.4 Vision Datasets Mixture

Following previous works (Ritter et al., 2018; Zeno et al., 2018), we perform continual learning on a sequence of 5 vision datasets: MNIST, notMNIST[1], FashionMNIST (Xiao et al., 2017), SVHN (Netzer et al., 2011) and CIFAR-10 (Krizhevsky, 2009) (see Appendix A.4 for dataset details). The MNIST, notMNIST and FashionMNIST datasets are zero-padded to be of size $32\times32$ and are replicated 3 times to create grayscale images with 3 channels, thus matching the resolution of the SVHN and CIFAR-10 images. Here, we use a CNN architecture that is similar to the one used in (Ritter et al., 2018; Zeno et al., 2018) (more details in Appendix A.4). The initial $\eta$ parameter value was set to 0.0001. We train the network with mini-batches of size 32 and optimized using plain SGD with a fixed learning rate of 0.01 for 50 epochs per task.

We found that DHP Softmax plus MAS decreases BWT by 0.04 resulting in a 2.14% improvement in average test accuracy over MAS on its own (see Table 1 and Appendix A.6, Figure 6). Also, SI with DHP Softmax outperforms other competitive methods with an average test performance of 81.75% and BWT of -0.04 after learning all five tasks. In Table 1, we present a summary of the final average test performance after learning all tasks in the respective continual learning problems. Here, we summarize the average test accuracy and BWT across ten trials for each of the benchmarks.

Table 1: The average test accuracy (%, higher is better) and backward transfer (BWT, lower is better) after learning all tasks on each benchmark, respectively. The results are averaged over 10 trials.

| Method | Permuted MNIST | Imbalanced Permuted MNIST | SplitMNIST | 5-Vision |
|---|---|---|---|---|
| Finetune | 76.73 / -0.19 | 76.44 / -0.20 | 90.43 / -0.13 | 60.02 / -0.33 |
| DHP Softmax | 78.49 / -0.16 | 80.85 / -0.14 | 98.23 / -0.02 | 62.94 / -0.26 |
| SI | 84.72 / -0.13 | 85.92 / -0.06 | 97.77 / -0.04 | 81.26 / -0.06 |
| DHP Softmax + SI | 85.20 / -0.09 | 85.39 / -0.06 | **99.15 / 0.00** | **81.75 / -0.04** |
| Online EWC | 86.24 / -0.11 | 87.18 / -0.09 | 97.65 / -0.03 | 78.61 / -0.07 |
| DHP Softmax + Online EWC | 87.30 / -0.09 | 87.43 / -0.08 | 98.96 / -0.01 | 79.10 / -0.04 |
| MAS | 88.52 / -0.08 | 87.32 / -0.09 | 98.24 / -0.02 | 78.51 / -0.05 |
| DHP Softmax + MAS | **89.53 / -0.06** | **88.80 / -0.05** | 98.43 / -0.01 | 80.66 / -0.01 |

## 5 Discussion and Conclusion

We have shown that the problem of catastrophic forgetting in continual learning environments can be alleviated by adding compressed episodic memory in the softmax layer through DHP and performing task-specific updates on synaptic parameters based on their individual importance for solving previously learned tasks. The compressed episodic memory allows new information to be learned in individual traces without overlapping representations, thus avoiding interference when added to the structured knowledge in the slow changing weights and allowing the model to generalize across experiences. The $\alpha$ parameter in the plastic component automatically learns to scale the magnitude of the plastic connections in the Hebbian traces, effectively choosing when to be less plastic (protect old knowledge) or more plastic (acquire new information quickly). The neural network with DHP Softmax showed noticeable improvement across all benchmarks when compared to a neural network with a traditional softmax layer that had an extra set of slow changing weights. The DHP Softmax does not introduce any additional hyperparameters since all of the structural parameters of the plastic part $\alpha$ and $\eta$ are learned, and setting the initial $\eta$ value required very little tuning effort.

---

[1] Originally published at `http://yaroslavvb.blogspot.com/2011/09/notmnist-dataset.html` and downloaded from `https://github.com/davidflanagan/notMNIST-to-MNIST`.

We demonstrated the flexibility of our model where, in addition to DHP Softmax, we can perform Hebbian Synaptic Consolidation by regularizing the slow weights using EWC, SI or MAS to improve a model's ability to alleviate catastrophic forgetting after sequentially learning a large number of tasks with limited model capacity. DHP Softmax combined with SI outperforms other consolidation methods on the Split MNIST and 5-Vision Datasets Mixture. The approach where we combine DHP Softmax and MAS consistently leads to overall superior results compared to other baseline methods on the Permuted MNIST and Imbalanced Permuted MNIST benchmarks. This is interesting because the local variant of MAS does compute the synaptic importance parameters of the slow weights $\theta_{i,j}$ layer by layer based on Hebb's rule, and therefore synaptic connections $i, j$ that are highly correlated would be considered more important for the given task than those connections that have less correlation. Furthermore, our model consistently exhibits lower negative BWT across all benchmarks, leading to higher average test accuracy over methods without DHP. This gives a strong indication that Hebbian plasticity enables neural networks to learn continually and remember distant memories, thus reducing catastrophic forgetting when learning from sequential datasets in dynamic environments. Furthermore, continual synaptic plasticity can play a key role in learning from limited labelled data while being able to adapt and scale at long timescales. We hope that our work will open new investigations into gradient descent optimized Hebbian consolidation for learning and memory in DNNs to enable continual learning.

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

# A DETAILS ON EXPERIMENTAL SETUP AND HYPERPARAMETER SETTINGS

In the continual learning setup, we train a neural network model on a sequence of tasks $T_{1:n_{\max}}$, where $n_{\max}$ is the maximum number of tasks the model is to learn in the respective benchmarks. Unlike the conventional supervised learning setup, continual learning trains a model on data that is fetched in sequential chunks enumerated by tasks. Therefore, in a continual learning sequence, the model receives a sequence of tasks $T_{1:n_{\max}}$ that is to be learned, each with its associated training data $(\mathcal{X}_n, \mathcal{Y}_n)$, where $\mathcal{X}_n$ is the input data and the corresponding label data denoted by $\mathcal{Y}_n$. Each task $T_n$ has its own task-specific loss $\mathcal{L}^n$, that will be combined with a regularizer loss term (refer to Eq. 4) to prevent catastrophic forgetting. After training is complete, the model will have learned an approximated mapping $f$ to the the true underlying function $\tilde{f}$. The learned $f$ maps a new input $\mathcal{X}$ to the target outputs $\mathcal{Y}_{1:n}$ for all $T_{1:n}$ tasks the network has learned so far. Also, it is to be noted that the set of classes contained in each task can be different from each other, as we have done in the SplitMNIST and Vision Datasets Mixture benchmarks. All experiments were run on either a Nvidia Titan V or a Nvidia RTX 2080 Ti.

## A.1 PERMUTED MNIST

We train the network on a sequence of tasks $T_{n=1:10}$ with mini-batches of size 64 and optimized using plain SGD with a learning rate of 0.01. We train for at least 10 epochs and perform early-stopping once the validation error does not improve for 5 epochs. If the validation error increases for more than 5 epochs, then we terminated the training on the task $T_n$, reset the network weights and Hebbian traces to the values that had the lowest test error, and proceeded to the next task.

**Hyperparameters:** For the Permuted MNIST experiments shown in Figure 2a, the regularization hyperparameter $\lambda$ for each of the task-specific consolidation methods is set to $\lambda = 100$ for Online EWC (Schwarz et al., 2018), $\lambda = 0.1$ for SI (Zenke et al., 2017b) and $\lambda = 0.1$ for MAS (Aljundi et al., 2018). We note that for the SI method, $\lambda$ refers to the parameter $c$ in the original work (Zenke et al., 2017b) but we use $\lambda$ to keep the notation consistent across other task-specific consolidation methods. In SI, the damping parameter, $\xi$, was set to 0.1. To find the best hyperparameter combination for each of these synaptic consolidation methods, we performed a grid search using a task sequence determined by a single seed. For Online EWC, we tested values of $\lambda \in \{10, 20, 50, \ldots, 400\}$, SI — $\lambda \in \{0.01, 0.05, \ldots, 0.5, 1.0\}$ and MAS — $\lambda \in \{0.01, 0.5, \ldots, 1.5, 2.0\}$.

## A.2 IMBALANCED PERMUTED MNIST

For each task in the Imbalanced Permuted MNIST problem, we artificially removed training samples from each class in the original MNIST dataset (LeCun et al., 2001) based on some random probability. For each class and each task, we draw a different removal probability from a standard uniform distribution $U(0, 1)$, and then remove each sample from that class with that probability. The distribution of classes in each dataset corresponding to tasks $T_{n=1:10}$ is given in Table 2.

Table 2: Distribution of classes in each imbalanced dataset for the respective tasks $T_{n=1:10}$.

| Classes | Tasks | | | | | | | | | |
|---|---|---|---|---|---|---|---|---|---|---|
| | 1 | 2 | 3 | 4 | 5 | 6 | 7 | 8 | 9 | 10 |
| 0 | 4459 | 3780 | 1847 | 3820 | 5867 | 122 | 1013 | 4608 | 908 | 3933 |
| 1 | 1872 | 3637 | 1316 | 6592 | 1934 | 1774 | 5533 | 2569 | 831 | 886 |
| 2 | 2391 | 4125 | 2434 | 4966 | 5245 | 4593 | 4834 | 4432 | 3207 | 3555 |
| 3 | 4433 | 1907 | 1682 | 278 | 3027 | 2315 | 5761 | 3293 | 2545 | 3749 |
| 4 | 186 | 2728 | 2002 | 151 | 1435 | 5829 | 1284 | 3910 | 4593 | 927 |
| 5 | 4292 | 2472 | 2924 | 1369 | 4094 | 4858 | 2265 | 3289 | 1134 | 1413 |
| 6 | 2339 | 3403 | 4771 | 5569 | 1414 | 2851 | 2921 | 4074 | 336 | 3993 |
| 7 | 4717 | 3090 | 4800 | 2574 | 4086 | 1065 | 3520 | 4705 | 5400 | 3650 |
| 8 | 3295 | 5493 | 76 | 4184 | 2034 | 4672 | 682 | 196 | 2409 | 1709 |
| 9 | 2625 | 3880 | 4735 | 1647 | 2645 | 3921 | 901 | 4546 | 4649 | 2045 |
| Total | 30609 | 34515 | 26587 | 31120 | 31781 | 32000 | 28714 | 35622 | 26012 | 25860 |

For the Imbalanced Permuted MNIST experiments shown in Figure 5, the regularization hyperparameter $\lambda$ for each of the task-specific consolidation methods is $\lambda = 400$ for Online EWC (Schwarz et al., 2018), $\lambda = 1.0$ for SI (Zenke et al., 2017b) and $\lambda = 0.1$ for MAS (Aljundi et al., 2018). In SI, the damping parameter, $\xi$, was set to 0.1. Similar to the Permuted MNIST benchmark, to find the best hyperparameter combination for each of these synaptic consolidation methods, we performed a grid search using a task sequence determined by a single seed. For Online EWC, we tested values of $\lambda \in \{50, 100,\ldots,1\times10^3\}$, SI — $\lambda \in \{0.1, 0.5,\ldots, 2.5, 3.0\}$ and MAS — $\lambda \in \{0.01, 0.05, \ldots, 1.5, 2.0\}$. Across all experiments, we maintained the the same random probabilities detemined by a single seed to artificially remove training samples from each class.

### A.3  SPLIT MNIST

**Hyperparameters:** For the Split MNIST experiments shown in Figure 2b, the regularization hyperparameter $\lambda$ for each of the task-specific consolidation methods is $\lambda = 400$ for Online EWC (Schwarz et al., 2018), $\lambda = 1.0$ for SI (Zenke et al., 2017b) and $\lambda = 1.5$ for MAS (Aljundi et al., 2018). In SI, the damping parameter, $\xi$, was set to 0.001. To find the best hyperparameter combination for each of these synaptic consolidation methods, we performed a grid search using the 5 task binary classification sequence (0/1, 2/3, 4/5, 6/7, 8/9). For Online EWC, we tested values of $\lambda \in \{1, 25, 50, 100, \ldots,1\times10^3, 2\times10^3\}$, SI — $\lambda \in \{0.1, 0.5, 1.0, \ldots, 5.0\}$ and MAS — $\lambda \in \{0.01, 0.05, 1.0,\ldots, 4.5, 5.0\}$. We train the network on a sequence of $T_{n=1:5}$ tasks with mini-batches of size 64 and optimized using plain SGD with a fixed learning rate of 0.01 for 10 epochs.

### A.4  VISION DATASETS MIXTURE

**Dataset Details:** The Vision Datasets Mixture benchmark consists of a sequence of 5 tasks where each task is a different image classification dataset: MNIST, notMNIST, FashionMNIST, SVHN and CIFAR-10. The notMNIST dataset consists of font glypyhs corresponding to letters 'A' to 'J'. The original dataset has 500,000 and 19,000 grayscale images of size $28\times28$ for training and testing, respectively. However, similar to MNIST, we only use 60,000 images for training and 10,000 for testing. FashionMNIST consists of 10 categories of various articles of clothing, and there are 60,000 and 10,000 grayscale images sized $28\times28$ for training and testing, respectively. SVHN consists of digits '0' to '9' from Google Street View images and there are 73,257 and 26,032 colour images of size $32\times32$ for training and testing, respectively. CIFAR-10 consists of 50,000 and 10,000 colour images of size $32\times32$ from 10 different categories for training and testing, respectively.

**Architecture:** The CNN architecture consists of 2 convolutional layers with 20 and 50 channels respectively, and a kernel size of 5. Each convolution layer is followed by LeakyReLU nonlinearities (negative threshold of 0.3) and $2\times2$ max-pooling operations with stride 2. The two convolutional layers are followed by an FC layer of size 500 before the final softmax output layer (refer to Table 3). Similar to (Ritter et al., 2018; Zeno et al., 2018), a multi-headed approach was used because the class definitions are different between datasets.

In the other benchmark problems, we use a single $\eta$ across all connections. In this benchmark, our model has a trainable $\eta$ value for each connection in the final output layer thus, $\eta \in \mathbb{R}^{m\times d}$ and we set the initial $\eta$ value to be 0.0001. We found that using separate $\eta$ parameters for each connection improved the stability of optimization and convergence to optimal test performance. This allows each plastic connection to modulate its own rate of plasticity when learning new experiences. It was observed that using a single $\eta$ value across all connections lead to instability of optimization on the SVHN and CIFAR-10 tasks.

**Hyperparameters:** For the 5-Vision Datasets Mixture experiments shown in Figure 6 the regularization hyperparameter $\lambda$ for each of the task-specific consolidation methods is $\lambda = 100$ for Online EWC (Schwarz et al., 2018), $\lambda = 0.1$ for SI (Zenke et al., 2017b) and $\lambda = 1.0$ for MAS (Aljundi et al., 2018). In SI, the damping parameter, $\xi$, was set to 0.1. To find the best hyperparameter combination for each of these synaptic consolidation methods, we performed a random search using the same task sequence ordering (MNIST, notMNIST, FashionMNIST, SVHN and CIFAR-10). For Online EWC, we tested values of $\lambda \in \{10, 50, 100,\ldots, 500\}$, SI — $\lambda \in \{0.01, 0.05, 0.1,\ldots, 1.0\}$ and MAS — $\lambda \in \{0.01, 0.05, 1.0,\ldots, 4.5, 5.0\}$.

Table 3: Network architecture used for Vision Datasets Mixture benchmark in Section 4.4. For convolutional layers, the output size denotes channel size of output. The negative threshold for all of the LeakyReLU nonlinearities were set to 0.2.

|  | output size | kernel | padding | stride |
|---|---|---|---|---|
| convolution | 20 | (5, 5) | (1,1) | (1,1) |
| LeakyReLU | - | - | - | - |
| MaxPool | - | - | - | (2, 2) |
| convolution | 50 | (5, 5) | (1, 1) | (1, 1) |
| LeakyReLU | - | - | - | - |
| MaxPool | - | - | - | (2, 2) |
| convolution | 50 | (5, 5) | (1, 1) | (1, 1) |
| LeakyReLU | - | - | - | - |
| MaxPool | - | - | - | (2, 2) |
| fully-connected | 500 | - | - | - |
| LeakyReLU | - | - | - | - |
| fully-connected | 10 | - | - | - |

## A.5 SENSITIVITY ANALYSIS

We provide a summary of the sensitivity analysis performed on the Hebb decay term $\eta$ and show its effect on the final average test performance after learning a sequence of tasks in the continual learning setup. The plots on the left and center in Figure 4 show the effect of the initial $\eta$ value on the final test performance after learning tasks $T_{n=1:10}$ in a sequential manner for the Permuted MNIST and Imbalanced Permuted MNIST benchmarks, respectively. We swept through a range of values $\eta \in \{0.1, 0.01, 0.001, 0.0005, 0.0001\}$ and found that setting $\eta$ to low values led to the best performance in terms of being able to alleviate catastrophic forgetting. Similarly, we also performed a sensitivity analysis on the $\eta$ parameter for the Split MNIST problem (see the rightmost plot in Figure 4). Table 4 presents the average test accuracy across 5 trials for the MNIST-variant benchmarks, which corresponds to the sensitivity analysis plots in Figure 4.

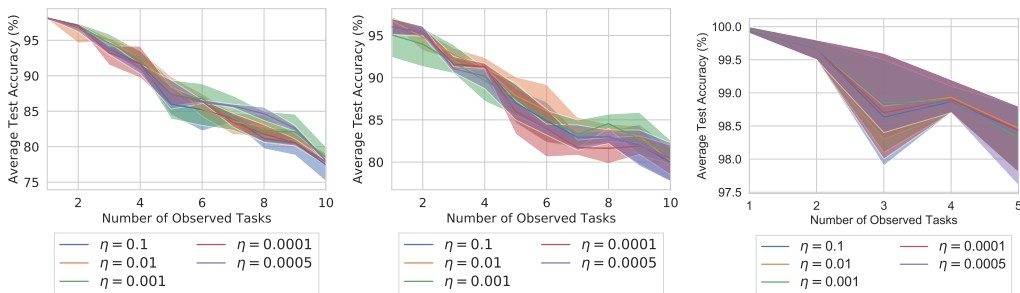

Figure 4: A sensitivity analysis on the Hebb decay term $\eta$ in Eq. 3. We show the average test accuracy for different initial values of $\eta$ after learning all tasks on the (left) Permuted MNIST, (center) Imbalanced Permuted MNIST and (right) Split MNIST problems. The shaded regions correspond to the standard error of mean (SEM) across 5 trials.

Table 4: The average test accuracy (%, higher is better) for different initial $\eta$ values after learning all tasks on the Permuted MNIST, Imbalanced Permuted MNIST and Split MNIST continual learning benchmarks, respectively. The results are averaged over 5 trials.

| Hebbian Plasticity Decay Term ($\eta$) | Permuted MNIST | Imbalanced Permuted MNIST | SplitMNIST |
|---|---|---|---|
| $\eta = 0.1$ | 77.43 | 80.00 | 98.43 |
| $\eta = 0.01$ | 78.60 | 80.13 | 98.47 |
| $\eta = 0.001$ | 78.49 | 80.85 | 98.32 |
| $\eta = 0.0001$ | 77.83 | 80.47 | 98.43 |
| $\eta = 0.0005$ | 78.05 | 80.40 | 98.37 |

## A.6 ADDITIONAL FIGURES FOR SPLITMNIST AND VISION DATASETS MIXTURE

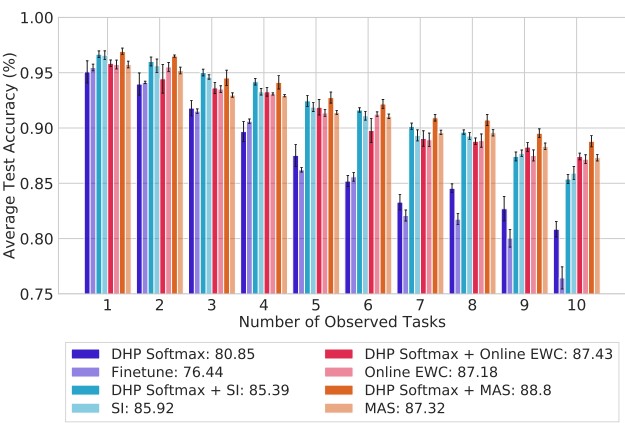

Figure 5: The average test accuracy on a sequence of on a sequence of 10 *imbalanced* Permuted MNIST tasks $T_{n=1:10}$. The average test accuracy over all learned tasks is provided in the legend. The shaded regions correspond to the SEM across 10 trials.

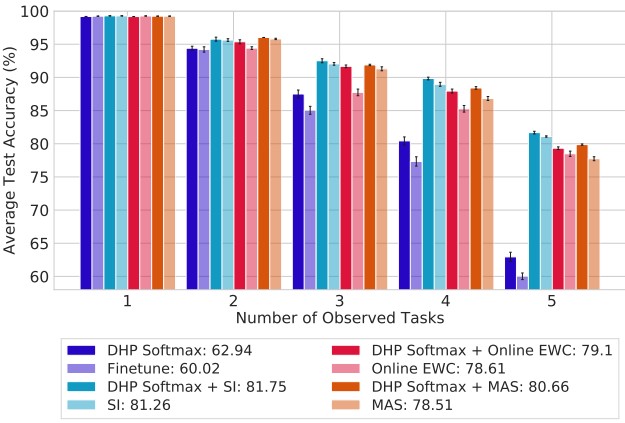

Figure 6: The average test accuracy on a sequence of 5 diffferent vision datasets $T_{n=1:5}$. The average test accuracy over all learned tasks is provided in the legend. The error bars correspond to the SEM across 10 trials.

## B  EXAMPLE PYTORCH IMPLEMENTATION OF DHP SOFTMAX LAYER

```python
class DHP_Softmax_Layer(nn.Module):
  def __init__(self, in_features, out_features, eta_rate=0.001):
    super(DHP_Softmax_Layer, self).__init__()
    """Applies a linear transformation to the hidden activations of the
    last hidden layer with an additional plastic component implemented
    using Differentiable Hebbian Plasticity (DHP):
    :math:'z = (w + \alpha * Hebb)h'.

    Args:
      in_features: size of each input in last hidden layer.
      out_features: number of classes.
      eta_rate: initial learning rate value of plastic connections.

    Returns:
      z: the softmax pre-activations (unnormalized log probabilities).
      hebb: the updated Hebbian traces for the next iteration.
    """
    self.in_features = in_features
    self.out_features = out_features
    self.eta_rate = eta_rate

    # Initialize fixed (slow) weights with He initialization.
    self.weight = Parameter(torch.Tensor(self.in_features,
                  self.out_features))
    init.kaiming_uniform_(self.weight, a=math.sqrt(5))

    # Initialize alpha scaling coefficients for plastic connections.
    self.alpha = Parameter((.01 * torch.rand(self.in_features,
                  self.out_features)),
                  requires_grad=True)

    # Initialize the learning rate of plastic connections.
    self.eta = Parameter((self.eta_rate * torch.ones(1)),
                  requires_grad=True)

  def forward(self, h, y, hebb):
    if self.training:
      for _, c in enumerate(torch.unique(y)):
        # Get indices of corresponding class, c, in y.
        y_c_idx = (y == c).nonzero()
        # Count total occurences of corresponding class, c in y.
        s = torch.sum(y == c)

        if s > 0:
          # Perform Hebbian update (lines 6-7 in Algorithm 1)
          h_bar = torch.div(torch.sum(h[y_c_idx], 0),
                    s.item())
          hebb[:,c] = torch.add(torch.mul(torch.sub(1, self.eta),
                      hebb[:,c].clone()),
                      torch.mul(h_bar, self.eta))

    # Compute softmax pre-activations with plastic (fast) weights.
    z = torch.mm(h, self.weight + torch.mul(self.alpha, hebb))

    return z, hebb

  def initial_zero_hebb(self):
    return Variable(torch.zeros(self.in_features, self.out_features),
```

57            r e q u i r e s _ g r a d = F a l s e )

Listing 1: PyTorch implementation of the DHP Softmax model which adds a compressed episodic memory to the final output layer of a neural network through plastic connections as described in Algorithm 1. We want to emphasize the simplicity of implementation using popular ML frameworks.

## C   DIFFERENTIABLE HEBBIAN CONSOLIDATION

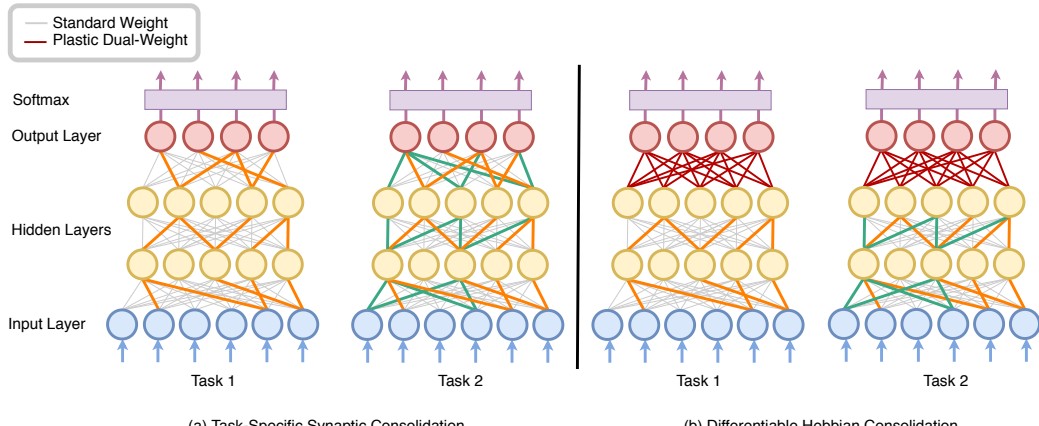

(a) Task-Specific Synaptic Consolidation          (b) Differentiable Hebbian Consolidation

Figure 7: The difference between (a) task-specific consolidation methods (e.g., EWC, SI and MAS) and (b) Differentiable Hebbian Consolidation in a simple two tasks continual learning scenario. The baseline standard weight connections are represented by light gray lines and the dual-weight plastic connections are represented by red lines. After learning Task 1, the parts of the network which are deemed important for Task 1 are indicated in orange. Task 2 has a different set of input patterns and utilizes the parts shown in green. The Differentiable Hebbian Consolidation framework consolidates representations learned by the network and the hidden activations are accumulated directly into the softmax output layer parameters when a class has been seen by the network. This allows the network to retain these learned deep representations for a longer timescale without having to sacrifice as much capacity by dynamically learning to adjust the degree of plasticity in the weights.

# D ADDITIONAL EXPERIMENTS

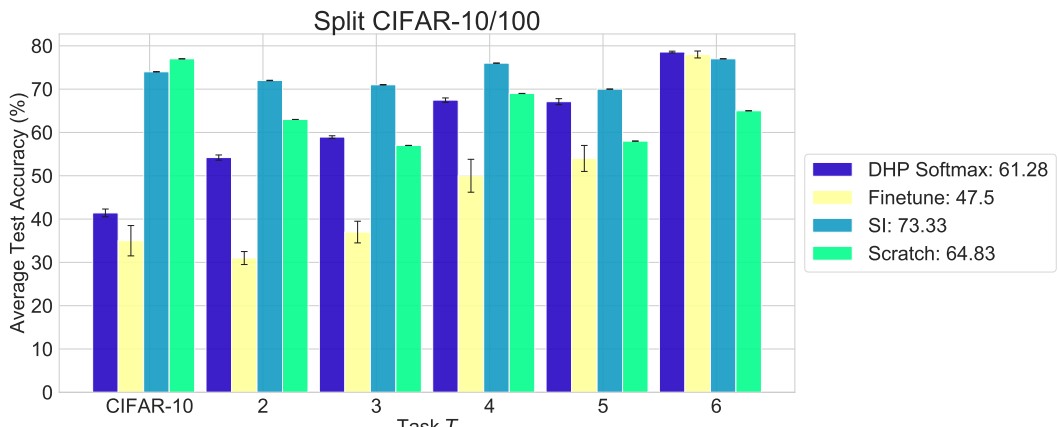

Figure 8: This replicates the Split CIFAR-10/100 experiment of Zenke et al. (2017b). First, the network was trained on the full CIFAR-10 dataset (Task $T_{n=1}$) and sequentially on 5 additional tasks each corresponding to 10 consecutive classes from the CIFAR-100 dataset (Tasks $T_{n=2:6}$). The test accuracies of CIFAR-10 and the CIFAR-100 splits are reported after having learned the final task in this sequence. The DHP Softmax (purple) alone significantly outperforms Finetune (yellow) on each of the tasks in this class-incremental learning setup. On some tasks, DHP Softmax alone performs as well or better than when training from scratch (light green). The test accuracies of Finetune, when training from scratch and SI (turquoise) were taken from von Oswald et al. (2019).

