# OpenReview forum: "Differentiable Hebbian Consolidation for Continual Learning"
_ICLR.cc/2020/Conference — Reject_

### Official Review · AnonReviewer1 · 2019-10-05
**Official Blind Review #1**

**Rating:** 3

**Review:**

This paper addresses the continual learning setting, and aims to mitigate catastrophic forgetting, with results on Permuted MNIST, Split MNIST, Vision Datasets Mixture, and their own class-imbalanced version of the Permuted MNIST dataset. The authors propose to augment differentiable plastic weights - a general neural network component - with class-specific updates (similarly to prior work, such as the Hebbian softmax) at the final layer of a neural network, prior to a softmax. While well-motivated in terms of the background and methodology (indeed, this is a simple way to prevent interference in fast weights), and nicely explored experimentally with lots of examinations into the workings of the method, the weak results on the simpler continual learning settings lead me to consider this a weak reject.

The authors show that fast weights can be applied in the continual learning setting, but alone they do not perform that well on the more challenging datasets, with mixed results on how much better they are as compared to a naive fine-tuning baseline, and they definitely lag behind synaptic consolidation methods. The authors' method combined with synaptic consolidation methods perform the best, but not much beyond the effect of the synaptic consolidation methods themselves. The authors are encouraged to evaluate their method with a large amount of classes, e.g. as done by iCaRL with CIFAR-100 and ILSVRC, with class-incremental training, to show if their method (a) scales (which I would anticipate, given the class-conditional Hebbian update) and (b) can deal with an alternative continual learning setting; a further resource is the work done around CORe50, which I would consider encapsulates more current thinking and practices around continual learning.

**Experience Assessment:**

I have read many papers in this area.

**Review Assessment: Checking Correctness Of Derivations And Theory:**

N/A

**Review Assessment: Checking Correctness Of Experiments:**

I carefully checked the experiments.

**Review Assessment: Thoroughness In Paper Reading:**

I read the paper thoroughly.

---

> ### Author Response · Authors · 2019-11-15
> **Response - additional experiments**
>
> We thank Reviewer 1 very much for carefully reading our paper and positive comments on our method and experimental examinations. We answer your question regarding additional experiments on class-incremental learning below:
>
> Re: 1. "The authors are encouraged to evaluate their method with a large amount of classes, e.g. as done by iCaRL with CIFAR-100 and ILSVRC, with class-incremental training, to show if their method (a) scales (which I would anticipate, given the class-conditional Hebbian update) and (b) can deal with an alternative continual learning setting; a further resource is the work done around CORe50, which I would consider encapsulates more current thinking and practices around continual learning."
>
> We would like to first emphasize that class-incremental learning methods such as iCaRL maintain exemplar patterns for each observed class and rehearse the model via distillation. Other techniques along these lines employ exact replay or generative replay. Exact replay strategies require storage of the data from previous tasks which are later replayed when learning new tasks. Generative replay strategies train a separate generative model to generate images to be replayed. For these reasons, these replay- or rehearsal-based techniques tend to perform very well in the class-incremental learning scenario. In our work, we focus on task-incremental continual classification, where tasks arrive in a batch-like fashion, and have clear boundaries (as examined through a large body of existing work in the recent review from De Lange et al., (2019) [1]). While we did not explicitly consider rehearsal techniques in our experiments for comparison, we attempt to address your question by evaluating DHP Softmax on the CIFAR-10/100 benchmark introduced by Zenke et al. (2017). Here, we compare our method against finetuning, training from scratch and synaptic intelligence (SI). DHP Softmax outperforms naive finetuning and even exhibits forward transfer of information, thus performing better than learning from scratch on some tasks. We added the additional experiments on Split CIFAR-10/100 to Appendix D of the paper. This experiment showcases the memory capabilities of DHP Softmax when learning new classes consecutively from the CIFAR-100 dataset. While we agree with Reviewer 1 that it would be interesting to evaluate our method with a large amount of classes (e.g., ILSVRC) or CORe50, we have yet to perform these experiments given the limited amount of time for experimentation during the rebuttal phase.
>
> [1] Matthias De Lange, Rahaf Aljundi, Marc Masana, Sarah Parisot, Xu Jia, Ales Leonardis, Gregory G. Slabaugh, and Tinne Tuytelaars. Continual learning: A comparative study on how to defy forgetting in classification tasks. CoRR, abs/1909.08383, 2019

---

> > ### Comment · AnonReviewer1 · 2019-11-15
> > **Additional Experiments**
> >
> > Your observation on this set of class-incremental methods like iCaRL is valid, and would be good to include in the paper integrated with the new text in the related work section. For Split CIFAR-10/100, overall SI performs very well across all splits after continual training, vastly outperforming DHP Softmax on nearly every split until the very end. On the final split, DHP Softmax, finetuning and SI all perform roughly the same. While I sympathise with the amount of time available during the rebuttal, I would like to see a stronger empirical evaluation. Alternative methods such as this work are important to develop, but given how they perform against or even in combination with other methods on these datasets, it remains unconvincing.

---

> > > ### Author Response · Authors · 2019-11-15
> > > **CLS methods in Related Work and Additional Experiments**
> > >
> > > Thank you Reviewer 1 for the quick response. We also thank you for validating our observations on this set of class-incremental methods like iCaRL. We would like to acknowledge that in the related work section of the paper, we did include CLS theory inspired strategies based on pseudo-rehearsal, episodic/exact replay, and generative replay. This can be found in the line "Our work draws inspiration from CLS theory which is a powerful computational framework for representing memories with a dual memory system via the neocortex and hippocampus. There have been numerous approaches based on CLS principles involving pseudo-rehersal (Robins, 1995; Ans et al., 2004; Atkinson et al., 2018), episodic replay (Lopez-Paz & Ranzato, 2017; Li & Hoiem, 2018) and generative replay (Shin et al., 2017; Wu et al., 2018)." We also mention that in our work, we are mainly interested in neuroplasticity-inspired CLS techniques for alleviating the catastrophic forgetting problem, where the approaches involving slow and fast weights is analogous to properties of CLS theory. To make things more clear, we have added more formal descriptions of replay- or rehearsal-based techniques in the related work section, as well as an additional reference to iCaRL which would be considered a hybrid technique due to the use of a fixed external memory and a distillation step (that overlaps with the regularization category).
> > >
> > > It also to be noted that on Split CIFAR-10/100, the DHP Softmax vastly outperforms fine-tuning and in some cases learning from scratch on each of the splits. Although SI alone outperforms DHP Softmax alone on every split except for the final split, we strongly believe that when DHP Softmax is used in-conjuction with SI or other task-specific synaptic consolidation methods, it will outperform SI on its own -- consistent with our other benchmark results in the paper on PermutedMNIST, Imbalanced PermutedMNIST, SplitMNIST and 5-Vision.  We would also like to emphasize that the 5-Vision benchmark involves having to continually learn from 5 different computer vision datasets in a sequential manner. This can be a very challenging problem for neural networks with fixed capacity to have to tackle catastrophic forgetting, and DHP Softmax alone shows noticeable improvement, as well as in-conjunction with other synaptic consolidation methods. We also believe that this paper puts forth an interesting alternative direction of research for continual learning in the context of "continual synaptic plasticity" which involves the dynamic modification of individual synaptic connections in neural networks to help address the overcoming catastrophic forgetting problem.

---

### Official Review · AnonReviewer2 · 2019-10-22
**Official Blind Review #2**

**Rating:** 6

**Review:**

The authors introduce DIFFERENTIABLE HEBBIAN CONSOLIDATION,a new framework for continual learning that can be implemented in the usual differentiable programming setups. This framework is motivated in terms of complementary learning system (CLS) theory which features an episodic memory module. The method is shown to be easily implemented as seen in their pytorch pseudocode (authors also suggest code will be released). Additionally, authors show the method leads to significant  improvements over simple baselines, and can complement other task-specific hebbian-based learning paradigms


As an alien to the continual learning literature, my judgment is mostly positive. First, the paper is well-written, and the method is well-motivated in terms of cognitive principles with a nice commentary about underlying neurobiological substrates. The description of alternative methods seems exhaustive (but see below). The resulting algorithm is simple and doesn't seem to entail a significant computational burden. Importantly, it is differentially programing-friendly. Results show improvements over a baseline. However, I have two significant criticisms regarding the experimental section that I hope the authors will address.

1)Authors comment on two paradigms to hebbian-based continual learning: the task-specific and theirs, based on CLS. Authors show their method complement improvements due to task-specific approaches (which is great), but proper comparisons to other CLS-based approaches seem missing. In other words, I would hope a much better statement than "
However, all of these methods were focused on rapid learning on simple tasks or metalearning over a distribution of tasks or datasets. Furthermore, they did not examine learning a large
number of new tasks while, alleviating catastrophic forgetting in continual learning" Perhaps the authors may try find experimental cases where comparison to such approaches are possible, and show the empirical results? I am concerned the finetuned MLP baseline could be a bit weak.
2)The authors should try more experiments. They showed many MNIST variants and a 5-vision dataset. However, results in other papers in the literature are shown in much more sophisticated contexts (e.g. learning to play games). I understand this could exceed the computational capabilities of authors, but  why, for example, not to try on a different dataset, as CIFAR? (as the Zenke et al paper did).

MInor: On section 3, page 4 authors mention the number of labels d but that number appears undefined

**Experience Assessment:**

I do not know much about this area.

**Review Assessment: Checking Correctness Of Derivations And Theory:**

I assessed the sensibility of the derivations and theory.

**Review Assessment: Checking Correctness Of Experiments:**

I assessed the sensibility of the experiments.

**Review Assessment: Thoroughness In Paper Reading:**

I read the paper at least twice and used my best judgement in assessing the paper.

---

> ### Author Response · Authors · 2019-11-15
> **Response (part 1) - evaluation of CLS-based neuroplasticity methods on continual learning**
>
> We appreciate the thoughtful questions from Reviewer 2, positive comments on our method and we are delighted to learn that Reviewer 2 feels that our paper is well written. We answer your questions below:
>
> Re: 1. "Authors show their method complement improvements due to task-specific approaches (which is great), but proper comparisons to other CLS-based approaches seem missing. In other words, I would hope a much better statement than ”However, all of these methods were focused on rapid learning on simple tasks or meta-learning over a distribution of tasks or datasets. Furthermore, they did not examine learning a large number of new tasks while, alleviating catastrophic forgetting in continual learning" Perhaps the authors may try find experimental cases where comparison to such approaches are possible, and show the empirical results? I am concerned the finetuned MLP baseline could be a bit weak."
>
> We agree that this statement is vague so we added a more detailed explanation on the extension of CLS-based neuroplasticity techniques for continual learning at the end of Section 2. There are several works on neural network models with fast weights as outlined in our paper and to our knowledge, we believe we are the first to propose a slow and fast weights framework for continual learning through the Differentiable Hebbian Consolidation framework.
>
> Out of the past works on CLS, the one that is closest to our work is the Hebbian Softmax layer [1]. The Hebbian Softmax modifies the parameters by an interpolation between Hebbian-learning and SGD updates. The authors’ motivation was to achieve fast binding for rarer classes, especially in the language-modelling setting where they show strong results. Our original intuition was that this would not work well in continual learning setups because this approach relies on an engineered scheduling scheme for annealing between Hebbian plastic and slow fixed weights to enable slow and fast weight updates. The scheduling scheme is designed with a class counter which counts class occurrences and is used for an annealing function. The Hebbian Softmax also has a smoothing limit which is the number of class occurrences before switching completely to SGD updates. The issue here is that if the same classes are observed frequently during training on a task, as in the case for the established continual learning benchmarks (i.e., PermutedMNIST, SplitMNIST, 5-Vision and etc.), the algorithm anneals to mainly SGD updates. Thus, the effect of the fast weights memory storage becomes non-existent and Hebbian Softmax eventually performs like a traditional softmax (i.e., finetuning) in continual learning setups.
>
> We confirmed our intuition by reimplementing the Hebbian Softmax and evaluating it on the same continual learning benchmarks considered for DHP Softmax. We found that DHP Softmax outperforms Hebbian Softmax in all of these benchmarks. Also, other methods such as meta networks [2, 3] combine fast weights based on non-trainable Hebbian learning with regular (slow) weights as well. However, their model was designed for meta-learning and performs very well on one-shot and few-shot learning benchmarks. We felt it was unfair to report and compare against fast weights models that were not designed for continual learning given some of their limitations. In addition to the Finetune baseline, we also evaluate EWC, SI and MAS without differentiable Hebbian plasticity as baselines. With a focus on continual learning, the goal of our work is to metalearn a local learning rule for the fast weights via the fixed (slow) weights and an SGD optimizer.
>
> We have improved the manuscript and provided some more discussion on why other neuroplasticity techniques inspired from CLS theory were not considered as baselines in the continual learning benchmarks we experimented on.
>
>
> [1] Jack W. Rae, Chris Dyer, Peter Dayan, and Timothy P. Lillicrap. Fast parametric learning with activation memorization. In Proceedings of the 35th International Conference on Machine Learning (ICML), pp. 4225–4234, 2018.
> [2] Tsendsuren Munkhdalai and Hong Yu. Meta networks. In Proceedings of the 34th International Conference on Machine Learning (ICML), pp. 2554–2563, 2017.
> [3] Tsendsuren Munkhdalai and Adam Trischler. Metalearning with hebbian fast weights. CoRR, abs/1807.05076, 2018.

---

> > ### Author Response · Authors · 2019-11-15
> > **Response (part 2) - additional experiments and notation clarification**
> >
> > Re: 2. "The authors should try more experiments. They showed many MNIST variants and a 5-vision dataset. However, results in other papers in the literature are shown in much more sophisticated contexts (e.g. learning to play games). I understand this could exceed the computational capabilities of authors, but  why, for example, not to try on a different dataset, as CIFAR? (as the Zenke et al paper did)."
> >
> > We agree that there have been other methods in the literature which have also been applied to reinforcement learning problems (e.g., learning to play games [1], maze exploration [2], and etc.). However, there are also a number of methods in the recent literature [3, 4, 5, 6] which have evaluated exclusively on the same vision benchmarks we considered (e.g., PermutedMNIST, SplitMNIST, 5-Vision Datasets Mixture). We would also like to emphasize that the 5-Vision benchmark involves having to sequentially learn from 5 different computer vision datasets. This can be a difficult problem for fixed capacity neural network architectures when having to tackle the catastrophic forgetting problem, and DHP Softmax shows noticeable improvement on its own and in-conjunction with other task-specific synaptic consolidation methods.
> >
> > We did our best to address Reviewer 2’s concerns by evaluating DHP Softmax on the Split CIFAR-10/100 benchmark introduced by Zenke et al. (2017), which involves training the network on the full CIFAR-10 dataset and then sequentially training on 5 consecutive CIFAR-100 splits, each corresponding to 10 classes from the CIFAR-100 dataset. We followed the same experimental setup as [4] and used the CNN model of Zenke et al. (2017). Inspired by [4], we compare against: finetuning, training from scratch and synaptic intelligence (SI). We observe that DHP Softmax alone achieves an average test accuracy of 61.28% after learning all 6 tasks in the sequence, in comparison to 47.50% from finetuning. On some tasks DHP Softmax outperforms training from scratch, indicating that there is some forward transfer of information between certain tasks. Although SI performs the best, we wanted to demonstrate the memory retention capabilities of DHP Softmax similar to SplitMNIST experiments where DHP Softmax alone significantly outperformed finetuning. We included the additional Split CIFAR-10/100 experiments in Appendix D.
> >
> > [1] Matthew Riemer, Ignacio Cases, Robert Ajemian, Miao Liu, Irina Rish, Yuhai Tu, , and Gerald Tesauro. Learning to learn without forgetting by maximizing transfer and minimizing interference. In International Conference on Learning Representations (ICLR), 2019.
> > [2] Vincenzo Lomonaco, Karen Desai, Eugenio Culurciello, and Davide Maltoni. Continual reinforcement learning in 3d non-stationary environments. arXiv preprint arXiv:1905.10112, 2019.
> > [3] Rahaf Aljundi, Marcus Rohrbach, and Tinne Tuytelaars. Selfless sequential learning. In International Conference on Learning Representations (ICLR), 2019.
> > [4] Johannes von Oswald, Christian Henning, Joa ̃o Sacramento, and Benjamin F Grewe. Continual learning with hypernetworks. arXiv preprint arXiv:1906.00695, 2019.
> > [5] Hongjoon Ahn, Donggyu Lee, Sungmin Cha, and Taesup Moon. Uncertainty-based continual learning with adaptive regularization. In Advances in Neural Information Processing Systems 32. 2019.
> > [6] Xilai Li, Yingbo Zhou, Tianfu Wu, Richard Socher, and Caiming Xiong. Learn to grow: A continual structure learning framework for overcoming catastrophic forgetting. In Proceedings of the 36th International Conference on Machine Learning (ICML), pp. 3925–3934, 2019.
> >
> > Re: 3. "On section 3, page 4 authors mention the number of labels d but that number appears undefined."
> >
> > We defined $d$ as the number of outputs of the last layer and we have provided a comment on the updated manuscript to clarify this.

---

### Official Review · AnonReviewer3 · 2019-10-24
**Official Blind Review #3**

**Rating:** 6

**Review:**

This paper tackles the problems of continual learning and catastrophic forgetting in neural networks. It uses methods inspired by Hebbia learning and complementary learning system (CLS), where there are slow and fast weights in the model.

The paper addresses an important topic. I find the idea of fast/slow weights to be a refreshing and different approach compared to previous work on catastrophic forgetting.

I had trouble following the details of the DHS Softmax, however. This may stem from my inexperience with Hebbian Learning, but here are some questions/suggestions:

- I am a bit confused exactly what is referred to as post/pre-synaptic connection, what is penultimate layer, etc. A figure might be helpful.

- It also might be helpful to write out an equation for the standard Softmax so it can be compared to Eq 2.

- Related to above, I am confused what is indexed by i,j in Eq 4. Compared to Eq 1, where theta only has one index (k), in Eq 4 theta has two indices (i,j).

- In terms of motivation, can you explain why this Hebbian strategy is applied only to the final softmax?

Other questions:

- Does it seem like DHS Softmax is not as strong by itself but works best in conjunction with others, such as EWC? I do not quite follow how they complement each other intuitively.

- Are there any hyperparameters required for DHS Softmax? It seems to be no?



**Experience Assessment:**

I do not know much about this area.

**Review Assessment: Checking Correctness Of Derivations And Theory:**

I did not assess the derivations or theory.

**Review Assessment: Checking Correctness Of Experiments:**

I did not assess the experiments.

**Review Assessment: Thoroughness In Paper Reading:**

I read the paper at least twice and used my best judgement in assessing the paper.

---

> ### Author Response · Authors · 2019-11-15
> **Response (part 1) - terminology, notations, design decisions and misc clarifications**
>
> Thank you Reviewer 3 for the positive comments about our approach and thoughtful questions. As you'll see below, they lead to a meaningful improvement in the framing of our paper. We have provided clarification on some terminology and notation, as well as more intuition on a couple of design decisions of our method. We answer your questions below:
>
> Re: 1. "I am a bit confused exactly what is referred to as post/pre-synaptic connection, what is penultimate layer, etc. A figure might be helpful."
>
> We apologize for the confusion. The terms pre/post-synaptic connection are now corrected to avoid ambiguity and are actually referring to the pre/post-synaptic neurons. The terms presynaptic neuron and postsynaptic neuron are often used when describing Hebbian plasticity, where the synaptic connection between two neurons $i$ and $j$ changes depending on the co-occurrence of pre- and postsynaptic activity. We employ a form of Hebbian plasticity which is the running average of the product between presynaptic activity and postsynaptic activity. Moreover, the connection between any two neurons $i$ and $j$ has both a fixed component and a plastic component (Eq. 2). We wanted to keep the terminology consistent with the literature on differentiable plasticity [1, 2, 3] and Hebbian learning-based plasticity rules [4, 5]. The penultimate layer here refers to the final fully-connected hidden layer of the neural network. We have added a figure in Appendix C comparing a standard neural network architecture with only task-specific synaptic consolidation and our proposed neural network architecture with Differentiable Hebbian Consolidation.
>
> [1] Thomas Miconi. Learning to learn with backpropagation of hebbian plasticity. CoRR, abs/1609.02228, 2016.
> [2] Thomas Miconi, Kenneth O. Stanley, and Jeff Clune. Differentiable plasticity: training plastic neural networks with backpropagation. In Proceedings of the 35th International Conference on Machine Learning (ICML), pp. 3556–3565, 2018.
> [3] Thomas Miconi, Aditya Rawal, Jeff Clune, and Kenneth O. Stanley. Backpropamine: training self-modifying neural networks with differentiable neuromodulated plasticity. In International Conference on Learning Representations (ICLR), 2019.
> [4] D. O. Hebb. The organization of behavior; a neuropsychological theory. Wiley, Oxford, England, 1949.
> [5] Terrence J. Sejnowski and Gerald Tesauro. The hebb rule for synaptic plasticity: Algorithms and implementations. 1989.
>
> Re: 2. "It also might be helpful to write out an equation for the standard Softmax so it can be compared to Eq 2."
>
> The standard softmax (baseline) would not include the plastic component, therefore Eq.2 can simply be written as $z_{j} =  \sum_{i=1}^{m} \theta_{i,j}$. In standard neural networks the weight connection has only fixed (slow) weights. Therefore, in Eq. 2, this is equivalent to setting a connection to be fully fixed (i.e., $\alpha$ = 0), hence removing the plastic component. Because Eq. 2 can reduce to the standard Softmax, we would prefer not to introduce another equation, but we have clarified this point in the text.
>
> Re: 3. "Related to above, I am confused what is indexed by i,j in Eq 4. Compared to Eq 1, where theta only has one index (k), in Eq 4 theta has two indices (i,j)."
>
> We apologize for the confusion and want to clarify that the index $k$ in Eq. 1 and $i,j$ in Eq. 4 both refer to the synaptic weight. We used a linear index for a general parameter in Eq. 1 as that is the typical notation in the continual learning literature. However, when we began to talk about synaptic connections in Section 3, we made the change from $k$ to $i,j$ to explicitly represent the weight of the connection from neuron $i$ to neuron $j$. As mentioned in our response to (1), we did this to keep the notation consistent with past works on differentiable plasticity and Hebbian learning-based plasticity rules. Thus, at any time, the total, effective weight of the connection between neurons $i$ and $j$ is the sum of a fixed (slow) component ($\theta_{i,j}$), plus a plastic (fast) component composed of: the Hebbian trace $Hebb_{i,j}$ multiplied by the plasticity coefficient $\alpha_{i,j}$.

---

> > ### Author Response · Authors · 2019-11-15
> > **Response (part 2)**
> >
> > Re: 4. "In terms of motivation, can you explain why this Hebbian strategy is applied only to the final softmax?"
> >
> > We apply the Hebbian strategy to the final softmax layer because our model explores an optimization scheme where hidden activations from deep representations extracted by the network are accumulated directly into the softmax output layer weights when a class has been seen by the network. This results in better representations in the early stages of learning and can also retain these learned deep representations for a longer timescale. Task-level fast learning, enabled by a highly plastic weight component, are updated within the scope of each task. Between tasks, this plastic component decays to prevent interference. But selective consolidation into a stable component protects old memories, effectively modelling plasticity over a range of timescales to form a learned neural memory (see Section 4.1). In Figure 3, Section 4.1 we observe that the Frobenius norm of the $\alpha$ grows, indicating that the network leverages the Hebbian traces as new tasks are learned in a sequential manner. Thus, by applying the Hebbian learning strategy to the final layer, we enable the network to consolidate a minimum number of synaptic weights while still being able to change enough synaptic weights if required by new task learning.
> >
> > Re: 5. "Does it seem like DHS Softmax is not as strong by itself but works best in conjunction with others, such as EWC? I do not quite follow how they complement each other intuitively."
> >
> > Task-specific synaptic consolidation techniques such as EWC, SI and MAS use regularization to slow down learning on slow weights deemed important to previously learned tasks. This is accomplished by estimating the importance of each parameter online during training on a task or after having trained on a task. DHP Softmax aims to learn task-specific representations in the fast weights memory storage that are rehearsed or replayed during training thus enabling “reactivation” of long-term memory traces in the slow weights (standard weights) of the neural network. The Differentiable Hebbian Consolidation framework ultimately learns meta-level knowledge across tasks and transfers its inductive biases through fast parameterization. As a result, we have two complementary systems: one that allows for the gradual acquisition of structured knowledge (fixed slow weights), and another that enables rapid learning of the specifics of individual experiences (plastic fast weights). Thus, DHP Softmax works best in conjunction with task-specific consolidation techniques, although DHP Softmax alone does show noticeable improvement in test accuracy and backward transfer across all experiments (PermutedMNIST, Imbalanced PermutedMNIST, SplitMNIST and 5-Vision) over naive finetuning.
> >
> > Re: 6. "Are there any hyperparameters required for DHS Softmax? It seems to be no?"
> >
> > The only hyperparameter that DHP Softmax introduces is the initial value of the Hebbian learning rate $\eta$, which also behaves as a decay term. $\eta$ itself is a learned parameter of the network that is optimized by SGD. We spent little to no effort on tuning the initial value of this parameter and we perform a sensitivity analysis to show the effect of the initial value on the final average test accuracy after learning a sequence of tasks on the Permuted MNIST, Imbalanced Permuted MNIST and SplitMNIST continual learning benchmarks (see Appendix A.5).

---

### Decision · Program_Chairs · 2019-12-19

**Decision:**

Reject

**Comment:**

The reviewers agreed that this paper tackles an important problem, continual learning, with a method that is well motivated and interesting. The rebuttal was very helpful in terms of relating to other work. However, the empirical evaluation, while good, could be improved. In particular, it is not clear based on the evaluation to what extent more interesting continual learning problems can be tackled. We encourage the authors to continue pursuing this work.